# A Pilot Study: Hypertension, Endothelial Dysfunction and Retinal Microvasculature in Rheumatic Autoimmune Diseases

**DOI:** 10.3390/jcm10184067

**Published:** 2021-09-09

**Authors:** Ahmed Mahdy, Martin Stradner, Andreas Roessler, Bianca Brix, Angelika Lackner, Adam Salon, Nandu Goswami

**Affiliations:** 1Physiology Division, Otto Loewi Center of Research in Vascular Biology, Immunity and Inflammation, Medical University of Graz, Neue Stiftingtalstraße 6, 8010 Graz, Austria; ahmed.mahdy@stud.medunigraz.at (A.M.); andreas.roessler@medunigraz.at (A.R.); bianca.brix@medunigraz.at (B.B.); adam.salon@medunigraz.at (A.S.); 2Rheumatology and Immunology Department, Medical University of Graz, Auenbruggerplatz 15, 8036 Graz, Austria; martin.stradner@medunigraz.at (M.S.); angelika.lackner@medunigraz.at (A.L.)

**Keywords:** endothelial dysfunction, hypertension, ADMA, connective tissue autoimmune diseases, early sign, pulse wave velocity

## Abstract

Background: The etiology of autoimmune rheumatic diseases is unknown. Endothelial dysfunction and premature atherosclerosis are commonly seen in these patients. Atherosclerosis is considered one of the main causes of cardiovascular diseases. Hypertension is considered the most important traditional cardiovascular risk. This case-control study aimed to investigate the relationship between autoimmune diseases and cardiovascular risk. Methods: This study was carried out in patients with rheumatoid arthritis, RA (*n* = 10), primary Sjögren syndrome, PSS (*n* = 10), and healthy controls (*n* = 10). Mean blood pressure (MBP), systolic blood pressure (SBP), diastolic blood pressure (DBP), and pulse wave velocity (PWV, an indicator of arterial stiffness) were assessed via a Vicorder device. Asymmetric dimethylarginine (ADMA) was measured via ELISA. Retinal photos were taken via a CR-2 retinal camera, and retinal microvasculature analysis was carried out. T-tests were conducted to compare the disease and control groups. ANOVA and ANOVA—ANCOVA were also used for the correction of covariates. Results: A high prevalence of hypertension was seen in RA (80% of cases) and PSS (40% of cases) compared to controls (only 20% of cases). Significant changes were seen in MBP (RA 101 ± 11 mmHg; PSS 93 ± 10 mm Hg vs. controls 88 ± 7 mmHg, *p* = 0.010), SBP (148 ± 16 mmHg in RA vs. 135 ± 16 mmHg in PSS vs. 128 ± 11 mmHg in control group; *p* = 0.007), DBP (77 ± 8 mmHg in RA, 72 ± 8 mmHg in PSS vs. 67 ± 6 mmHg in control; *p* = 0.010 in RA compared to the controls). Patients with PSS showed no significant difference as compared to controls (MBP: *p* = 0.240, SBP: *p* = 0.340, DBP: *p* = 0.190). Increased plasma ADMA was seen in RA (0.45 ± 0.069 ng/mL) and PSS (0.43 ± 0.060 ng/mL) patients as compared to controls (0.38 ± 0.059 ng/mL). ADMA in RA vs. control was statistically significant (*p* = 0.022). However, no differences were seen in ADMA in PSS vs. controls. PWV and retinal microvasculature did not differ across the three groups. Conclusions: The prevalence of hypertension in our cohort was very high. Similarly, signs of endothelial dysfunction were seen in autoimmune rheumatic diseases. As hypertension and endothelial dysfunction are important contributing risk factors for cardiovascular diseases, the association of hypertension and endothelial dysfunction should be monitored closely in autoimmune diseases.

## 1. Background

Autoimmune diseases are diseases in which the immune system loses its tolerance to self-antigen and starts to attack its tissues and organs [1]. The causes of autoimmune diseases are not known, but many factors can predispose them, including genetic, environmental, food, and stress [2]. Autoimmune diseases affect about 5–7% of the population, and their incidence is on the rise, without any clear etiology [3]. Autoimmune diseases are characterized by a chronic state of immune-mediated inflammation, as well as disturbances in the equilibrium of chemical mediators in the body, which have consequences for several body systems and organs [4,5,6]. Dysfunction of the immune system can lead to recognizing self-antigens as “foreign” and is associated with the release of antibodies against self-antigens [1]. This interaction between self-antigens and antibodies results in complement activation, deposition of immune complexes in the wall of blood vessels, tissue injury, cell death, and, in some cases, even loss of organ function [7]. Rheumatoid arthritis and Sjögren syndrome are prototypical autoimmune diseases.

The endothelium is the thin innermost layer in the blood vessels. Endothelial cells secrete chemical mediators that can cause constriction and dilation of blood vessels and ensures smooth blood flow in vessels as part of the body’s homeostasis. One of the most important chemical mediators secreted by the endothelium is nitric oxide, which plays a major role in blood circulation. Nitric oxide is the main vasodilator of blood vessels and prevents atherosclerosis, platelet aggregation, leucocytes adhesion to the vessels wall, and thrombosis [8,9]. Disturbances in nitric oxide secretion or function may lead to endothelial dysfunction. As the endothelial cells become dysfunctional, they lose their function, which can initiate the atherosclerosis process. Endothelial dysfunction is an early step in the process of atherosclerosis. Endothelial dysfunction is reversible if recognized early and treated. If left unmanaged, it can lead to atherosclerosis and cardiovascular diseases [10].

Asymmetric dimethylarginine (ADMA) is a competitive inhibitor of nitric oxide synthases (NOSs), a group of enzymes that catalyze the production of nitric oxide (NO) from L-arginine. ADMA, by its antagonizing action on nitric oxide, plays an important part in the development of endothelial dysfunction [11,12]. ADMA is also considered a major renal toxin and has been implicated in the occurrence of greater cardiovascular risk in renal failure patients [13]. As arterial stiffness and premature atherosclerosis are common in autoimmune diseases, ADMA has been observed to be higher in these diseases [14,15,16]. ADMA is also considered an independent risk factor in cardiovascular diseases [17,18]. The role of endothelial dysfunction and nitric oxide in hypertension pathogenesis, however, is currently unclear [19].

Hypertension is considered the most important traditional cardiovascular risk factor in the development of atherosclerosis and cardiovascular diseases [20]. Up to 90% of hypertension is primary hypertension, which is of unknown etiology. Whether autoimmune diseases are associated with hypertension or not is currently not well established. Some studies have reported that hypertension prevalence in autoimmune diseases ranges from 3% to 70% [21,22,23,24,25,26,27,28], while others have reported no difference in blood pressure between diseases and control groups [29,30,31,32,33]. 

The relation between chronic inflammation in autoimmune diseases, endothelial dysfunction, and atherosclerosis is still unknown [34]. What roles traditional risk factors like age, male gender, hypertension, diabetes, smoking, obesity, dyslipidemia, diabetes, and sedentary lifestyle, as well as nontraditional risk factors like inflammation, autoantibodies, drugs, and inflammatory cells (B and T cells), play in the disease pathogenesis and progression, are currently unknown [35]. Many cytokines liberated during inflammation, like tumor necrosis factor (TNF) [36] and ILs [37], have a great role in the initiation and continuation of inflammation; these roles are in addition to immunocomplex depositions in the wall of blood vessels [38]. In addition to that, many antibodies like rheumatoid factor, anti-citrullinated peptide antibody, anti-nuclear antibodies (ANA) [39,40,41,42,43,44], and anti-low density lipoprotein antibodies are believed to play a role in atherosclerosis formation [45].

Retinal vessel diameter is affected by many factors like age, weight, height, and eye diseases, but the most important factor is blood pressure [46,47]. Usually, there is autoregulation in cerebral circulation, including the retina, so with an increase in blood pressure, there is vasoconstriction and a decrease in the diameter of the retinal blood vessel to keep blood flow to the brain constant [48]. 

Pulse wave velocity is a well-known technique to assess arterial stiffness [49,50]. It depends on the measurement of the transmission velocity of the arterial wall wave between two arterial points; the most common maneuver is measuring carotid femoral pulse wave velocity [51]. The normal range of PWV is about 6 to 13 m/s according to a person’s age, height, and blood pressure. Any disease that affects arterial wall elasticity will influence the stiffness of blood vessels and consequently the velocity. Atherosclerosis, by its association with vascular stiffness, has been shown to affect PWV [52].

Cardiovascular diseases are the endpoint of endothelial dysfunction, whether in autoimmune diseases or other diseases. The current European League Against Rheumatism (EULAR) guidelines state that the cardiovascular risk in autoimmune disease is 1.5× greater when compared to the cardiac risk in patients not having chronic inflammation [53]. Chronic inflammation has been reported to be an independent cardiovascular risk [14].

To our knowledge, this study is the first study to investigate the relationship between endothelial dysfunction and hypertension in rheumatoid arthritis and Sjögren disease as compared to age-matched healthy persons, as well as whether endothelial dysfunction is already present in autoimmune diseases, and the relation between endothelial dysfunction and hypertension, cause or result.

The following is a preliminary case-control study aimed to investigate the relationship between autoimmune disease and cardiovascular risk factors and to study the effect of chronic inflammation on the vascular function that utilized patients with two autoimmune diseases. The diseases studied are RA, as the prototype of rheumatological autoimmune diseases with excessive inflammation on vascular function, and PSS, with mild or moderate inflammation. The specific aim was to assess the effects of these diseases and use this data for bigger epidemiological studies, including with other diseases.

## 2. Material and Methods

### 2.1. Study Design and Population

A case-control pilot study was carried out at the Rheumatology Clinic at the Medical University of Graz, Austria, between February 2021 and March 2021. There were 3 groups of participants: patients with rheumatoid arthritis, primary Sjögren syndrome, and age-matched healthy controls.

### 2.2. Inclusion/Exclusion Criteria

Male or female adults above 18 years old who were diagnosed with rheumatoid arthritis by the 2010 ACR/EULAR classification criteria [54], and primary Sjögren syndrome in line with the 2016 ACR/EULAR classification criteria [55], attending the rheumatology clinic in the Medical University of Graz. The recruited participants were free from previous histories of cardiovascular or cerebrovascular diseases. Patients who were suffering from active disease, infection, malignancy, or were pregnant or breastfeeding were excluded. The healthy controls were age-matched and free from autoimmune diseases or cardiovascular diseases.

### 2.3. Ethical Approval and Data Storage

The study was conducted in accordance with the principles stated in the Declaration of Helsinki of 1964 and its most recent revision (2013), as well as local and national regulations in Graz, Austria. Ethical approval was obtained from the Ethical Committee at the Medical University of Graz (Project identification code: 32-283 ex 19/20). The patients were informed about the study aims and the detailed protocol. Written informed consent was obtained from participants before enrolling them in the study. 

The study adhered to the standards of reporting and was in accordance with the National Data Protection act, wherein each participant was assigned a code and data samples were stored anonymously.

### 2.4. Data Collection

Data related to age, gender, weight, height, diagnosis, medication, lifestyle factors such as smoking, alcohol consumption, or physical activity were recorded.

### 2.5. Blood Pressure Measurements

Blood pressure was measured in a quiet room using the Vicorder device (SMT medical GmbH & Co. KG, Würzburg, Germany). After resting for 10 min, the patients explained the maneuver and consent given. Then participants remained in a supine position for another 10 min prior to measurements. A slandered 7 cm pressure cuff was used to measure systolic blood pressure (SBP), diastolic blood pressure (DBP). Mean blood pressure (MBP) was calculated. Hypertension is defined by patients taking medication for hypertension or blood pressure over 140 systolic and 90 diastolic.

### 2.6. Pulse Wave Velocity

PWV measured by the arterial wall stiffness was assessed non-invasively by a Vicorder device. The Vicorder measures the speed of waves transmitted between 2 points through the wall of large carotid and femoral arteries. The device is connected to a laptop software program that analyzes the wave and gives the result of the speed in m/s. Following 20 min of supine relaxation, pulse wave velocity was assessed.

### 2.7. Retinal Microvasculature Imaging

This was carried out using the CR-2 Canon automated camera. The software program Mona-Riva was used for analyzing the diameter of the six biggest arterioles and six biggest venules in the retina. Assessed were the central retinal arteriole equivalent (CRAE) and central retinal venule equivalent (CRVE), as well as the A/V ratio (% of CRA to CRV). 

### 2.8. Blood Collection

10 cm of blood was obtained from the participants. Blood sample taken for measuring lipid profile (total cholesterol, high-density lipoprotein (HDL), low-density lipoprotein (LDL)), serum creatinine, and endothelial dysfunction markers as ADMA. ADMA was assessed by enzyme-linked immunosorbent assay (ELISA: Immunodiagnostic AG; Bensheim, Germany). The normal range for the kits was 0.29–63 ng/mL.

### 2.9. Data and Statistical Analysis

Statistical analysis was performed using the Statistical Package for Social Sciences (SPSS) software (version 26, SPSS Inc.; Chicago, IL, USA). Statistically, we used t-tests for comparison between the disease and control groups. ANOVA and ANOVA—ANCOVA was used for the correction of significant covariates (age, weight, and height). Additionally, the Pearson correlations of the parameters with each other were carried out. The descriptive statistics were given as a number, percentage, and mean ± standard deviation, and median (range). A *p* < 0.05 was considered as the level of statistical significance except where an alpha level correction was necessary (Benjamini-Hochberg).

Due to the limited sample size, multivariate models may not be appropriate. However, this pilot study will help us to plan for bigger epidemiological studies, in which we will use a general mixed model to assess these effects across the sexes, over repeated measurements, and across the participant groups.

Correlation coefficient (r) results between 1 and −1 with results ±1, indicates a linear correlation in the same or other direction. We took 0–0.3 to be no correlation, 0.4–0.6 as moderate correlation, and 0.7–1 as a significant correlation [56].

To reduce the inter- and intra-rate variability of vascular assessments, we ensured that all the vascular measurements were only carried out by one person each time and across all the patients.

## 3. Results

The patient’s database stored in the rheumatology department at the Medical University of Graz was examined. Disease activity was 3.5 (median) (clinical disease activity index (CDAI, score range 0–10) in RA and 3 (EULAR Sjögren syndrome disease activity index (ESSDAI, range 0–6) in PSS. All patients were stable during their follow-ups in outpatient clinics at the time of examination.

Each of the investigated groups consisted of 10 patients; in RA, there were eight females, two males, age 55.3 ± 7 years, 79 kg (57–140). Only four out of the eight patients were taking antihypertensive medication. As none of the patients were taking vasodilators or diuretics to control their blood pressure, we believe that our results could not have been influenced by the medications of the patients. All the groups were taking biological disease-modified anti-rheumatoid drugs (DMARDS) plus Methotrexate (7 of 10). PSS group (10 females, 52.2 ± 8 years, 57.5 kg (49–145)) patients were not known to be hypertensive, and only one patient received mycophenolate mofetil. The healthy control group included two males and eight females of age 50 ± 9 years and weight 72.5 kg (60–105). Table 1 below outlines details of each of the three groups. Only two PSS patients used steroids in low doses. While these medications control the disease activity, they do not directly influence blood pressure. All the patients were not taking anti hyperlipidemia drugs (which are also considered anti-atherosclerotic medications). None of the patients were on any anti-inflammatory medication.

In the RA group, 80% of the patients had elevated blood pressure (SBP: 148 ± 16 mmHg, DBP: 77 ± 8 mmHg, MBP 101 ± 11 mmHg) while 40% of PSS patients had elevated blood pressure (SBP: 135 ± 16 mmHg, DBP: 72 ± 8mmHg, MBP: 93 ± 10 mmHg). However, only 20% of the controls showed elevated blood pressure (SBP: 128 ± 10 mmHg, DBP: 68 ± 5 mmHg, and MBP: 88 ± 7 mmHg). Statistical analyses showed significant differences between the RA group vs. the control group (SBP, *p*-value: 0.007, DBP, *p*-value: 0.01, and MBP *p*-value: 0.01) (Table 2). PSS patients showed no significant difference as compared to the controls (SBP, *p*-value: 0.34, DBP, *p*-value: 0.19, MBP *p*-value: 0.24). However, significant values are seen when the autoimmune diseases are compared with the healthy controls (SBP, *p*-value: 0.02, DBP, *p*-value 0.01, MBP, *p*-value: 0.02). 

No correlation between elevated blood pressure and kidney function in regard to serum creatinine was seen (0.154). With regards to the age of the participants, there is a correlation between age and blood pressure in the diseased group (0.714) but not in the control group (0.408). 

Elevated ADMA levels were seen in the plasma of the diseased group. There was a significant difference between RA patients as compared to controls (RA: 0.45 µmol/L, ±0.06; controls: 0.38 µmol/L, ±0.05; *p* = 0.02). However, there were no statistical differences in the ADMA levels of PSS patients as compared to controls (Table 1). Similarly, no direct significant relationships were seen between ADMA levels and SBP (0.431), DBP (0.178), or MBP (0.321). No correlation between ADMA levels and disease duration was seen (0.06). There was a negative significant correlation between ADMA and CRAE in the disease group (PSS more than RA) (−0.517).

Age and ADMA: No correlations between age and ADMA levels were seen (0.13). On the other hand, there was a significant correlation between ADMA and weight (0.612) as well as creatinine level (disease group, not control group) (0.525). However, no significant correlation between ADMA and PWV (0.348) (only in PSS) was observed (0.669).

Pulse wave velocity values varied across the three groups: RA group (10.2 ± 2.31 m/s), PSS (8.74 ± 1.64 m/s), and control group (8.69 ± 1.78 m/s). However, no statistical differences were seen between the patient groups compared to the controls (Table 2). There is a significant correlation between blood pressure (systolic and mean) and pulse wave velocity in autoimmune disease groups and the control group. In the RA and PSS groups, statistically significant correlations between some blood pressure values and PWV were seen: SBP (*p* = 0.02, *p* = 0.003, respectively) and MBP (*p* = 0.04, *p* = 0.01, respectively) but not in DBP (*p* = 0.12, *p* = 0.09, respectively). The control group showed significant correlations in SBP (*p* = 0.02).

Age and pulse wave velocity: There was a significant correlation between age and PWV (arterial stiffness) in all the groups (r = 0.703); on the other hand, no correlation between PWV and disease duration was found (r = 0.179).

Analysis of retinal blood vessels showed no statistically significant differences between the diseased group and healthy controls. However, there was a decrease in retinal arterial diameter in the RA group (138 mic ± 6, *p* = 0.125) as compared to the PSS patients (146 mic ± 17, *p* = 0.58) and control group (151 mic ± 21). The retinal vein diameter showed no changes between all groups. 

A negative significant correlation between weight (−0.482), height (−0.578), ADMA (−0.517), and BP (−0.465) was seen with CRAE in the diseased group. This was, however, not the case in the healthy controls. Finally, no correlations between cholesterol level values and those of blood pressure (0.252), ADMA (−0.105), PWV (0.13), or CRAE (0.17) were seen.

## 4. Discussion

It was surprising to report that even with this limited number of participants in each group, we were able to see statistically significant differences across the populations studied. We also observed a high prevalence of hypertension in patients with the two types of autoimmune diseases. This result is particularly important, as currently, the evidence regarding the true prevalence of hypertension in autoimmune diseases is still not clear. Moreover, our findings also contribute to the ongoing debate regarding hypertension and vascular functional changes in autoimmune patients—is it hypertension that causes endothelial dysfunction, or does the endothelial dysfunction, which accompanies chronic inflammatory states, lead to hypertension. Our paper adds to the exciting and important discussions in this area. However, these data are going to be used for planning and carrying out bigger epidemiological studies in the future.

We also agree that a comparison between two diseases can be a problem, especially when a major, very common disease is compared with another less known disease. However, it was not our intention to compare RA and PSS; rather, we compared chronic inflammation in autoimmune disease patients and healthy persons. 

Our study showed that elevated blood pressure is a very common sign in autoimmune diseases, especially in rheumatoid arthritis. In addition, there appears to be an associated endothelial dysfunction, as shown by elevated plasma ADMA levels but no accompanying effects on macro- or micro-vascular circulation. Arterial stiffness, using pulse wave velocity measurements, was used to assess general vascular function, while the microvascular function was assessed via retinal imaging studies in which central retinal arteriole and central retinal venule diameters were measured. 

Currently, there is debate about the percentage of hypertension in autoimmune diseases. While some studies reported the occurrence of accompanying hypertension, others did not observe blood pressure differences between those with autoimmune diseases and healthy persons [21,29,33].

Our results show that hypertension prevalence is high in autoimmune diseases, especially in rheumatoid arthritis. From the foregoing discussion, it appears that the occurrence of hypertension in autoimmune diseases has been poorly studied and underreported. This is really alarming as it is important to assess and manage hypertension correctly. Moreover, there are currently no guidelines for the treatment of hypertension in autoimmune diseases. When hypertension is seen in autoimmune patients, most doctors only use the traditional antihypertensive drugs to control hypertension [57]. Therefore, our results are important and lay the foundation for further discussions and recommendations related to the management of hypertension in autoimmune diseases.

All studies have identified hypertension as one of the traditional cardiovascular risk factors affecting the atherosclerosis process and have compared traditional versus nontraditional risk factors as a cause of premature atherosclerosis. An interesting question put forward is whether hypertension associated with autoimmune diseases arises due to a chronic state of inflammation of blood vessels due to low-grade vasculitis, immune complex deposition (especially renal circulation, with loss of vasodilator mediators, and predominant vasoconstrictors, endothelial dysfunction), medications (e.g., steroid or nonsteroidal anti-inflammatory drugs), or whether it is a co-incidental finding [58,59]. Similarly, other currently unanswered questions include the following: Does premature atherosclerosis and arterial stiffness occur in these diseases due to traditional risk factors like age, male gender, hypertension, or hyperlipidemia? Does it arise from the autoimmune diseases themselves (e.g., due to accompanying inflammation and/or medications that these patients receive)? Or whether hypertension itself is a mechanical factor, which is responsible for endothelial injury and then endothelial dysfunction leading to loss of vasodilator ability of endothelium [60]? Finally, is it possible that the endothelial dysfunction produced by chronic inflammation in autoimmune diseases is responsible for hypertension, which then leads to premature atherosclerosis [61,62,63]? Several authors have tried to answer these questions with varying conclusions. For example, Konukoglu and Uzun concluded that it was not clear whether hypertension is the cause of inflammation or arises due to it [19]. Kim Lauper and colleagues concluded that atherosclerosis in autoimmune diseases arises due to chronic inflammatory processes which affect lipid metabolism and/or are involved in the pathology of premature atherosclerosis [28]. Our study showed a marked significant difference between elevated blood pressure in autoimmune disease patients as compared to controls. We observed a prevalence of hypertension in rheumatoid arthritis of 80% and in Primary Sjögren’s syndrome of 40% as compared to only 20% age and sex-matched healthy controls. Hypertension in our study was defined as systolic blood pressure higher than 140 mmHg and diastolic more than 90 mmHg. Similarly, patients with histories of antihypertensive treatments were also classified as hypertensive. Our results of the prevalence of hypertension in autoimmune diseases are much higher than what has been previously reported. For instance, Roman and his colleagues reported a prevalence of only 18% [31]. Han and his colleagues showed in a large cohort that the prevalence of hypertension in rheumatoid arthritis is 34% versus 23% in the controls [25]. Maxime Dougados and his colleagues reported a prevalence of hypertension in the rheumatoid arthritis group of 11% [22], while Roman and colleagues reported the prevalence as only 3% [28]. On the other hand, Chung and his colleagues reported values of up to 73% [23]. Others, such as Panoulas and his colleagues stated that hypertension is very prevalent but remains underdiagnosed and undertreated [21]. The differences in the percentage of hypertension in these studies may be related to the number of diseased patients recruited, the definition of hypertension used, and/or the ethnicity of the participants, diseases duration, or activity state of the diseases and time of measurements in relation to the relapse or remission of diseases. While some studies suggested that the increase in blood pressure in the rheumatoid group is due to age, not due to the disease itself [23,64,65], we found no correlation between age and blood pressure in the normal population. The results of our pilot study are significant, as they add to the literature regarding the prevalence of hypertension in autoimmune disease but also lays the foundation for further research, which can help in answering the question: what comes first? Is it hypertension that leads to endothelial dysfunction, or is hypertension due to the accompanying endothelial dysfunction in autoimmune diseases? Based on our pilot observational study data (hypertension prevalence and ADMA level), we hypothesize that hypertension seen in autoimmune diseases may be the result of endothelial dysfunction. Our hypothesis is supported by the results of Rossi and his colleagues [66] but not supported by the observations of Shimbo and his colleagues, who showed that hypertension leads to endothelial dysfunction [67]. Similarly, Taddei and his colleagues suggest that endothelial dysfunction is not a cause of hypertension [68]. Apparently, more studies are required to assess the processes that underlie the pathophysiology of vascular function changes that occur in autoimmune diseases.

With regards to pulse wave velocity changes in autoimmune diseases, our study showed that there was a tendency to have a higher PWV in rheumatoid patients than in healthy controls; however, it did not reach statistical significance. The patients who showed higher PWV also had hypertension, so it is not known whether this high PWV was due to the accompanying high inflammatory state in autoimmune patients or due to hypertension? Other studies have also reported increased arterial stiffness in RA patients [69]. Furthermore, Kocabay and his colleagues reported increased PWV in SLE, RA, and in Behçet’s disease [70]. In a meta-analysis by Wang and his colleagues on SLE patients, they found an increase in PWV [71]. Awalia and his colleagues speculated that the arterial stiffness seen in RA is related to the disease activity, not disease duration (greater the severity of the disease, greater the arterial stiffness) [72]. Dzieża-Grudnik and his colleagues found no difference in PWV in active short disease duration RA patients [73]. Although we did not find a difference between Primary Sjögren Syndrome patients and the control group, Sabio and colleagues found increased PWV in Sjögren patients [74]. Moreover, Atzeni and colleagues reported similar results [75]. Yong and his colleagues. carried out a meta-analysis for Primary Sjögren patients and concluded that higher PWV and arterial stiffness are seen in these patients [76]. We did not, however, observe any differences in PWVs in the primary Sjögren group (median disease duration: 5 years). The difference between our study and others may be that the other studies use different age groups (here, the participants were slightly younger) and different ethnicity or disease activity or duration.

With regards to the ADMA plasma levels, our results are similar to the other studies, which showed that ADMA is higher in rheumatologically autoimmune diseases [75]. Erre and his colleagues. carried out a meta-analysis and concluded that there is an elevated level of ADMA in autoimmune diseases [77].

Only a few studies have examined the retinal vessel diameters in autoimmune diseases, especially in rheumatoid arthritis and primary Sjögren syndrome. One study reported that there is an increase in CRVE in the rheumatoid group [78]. Similarly, Van Doornum and colleagues also observed an increase in venular diameter in rheumatoid diseases but not in CRAE [79]. We did not find significant differences between retinal vessel diameter (CRAE, CRVE) in the diseased groups, RA and PSS, compared to the control group. The CRAE in the RA group is, however, smaller than the diameter in the control group. Decreases in the diameter of CRAE that we observed in the rheumatoid group may be due to the prevalence of hypertension in this group rather than the disease itself. Future studies should examine in detail how retinal vessel diameter can be used to assess microvascular effects of these diseases [46].

Until now, there is still controversy about the percentage of hypertension in autoimmune diseases, and, as we mention in our article, the difference in the prevalence of hypertension is big. We found that nearly all the RA patients had hypertension, although they are young, and there was no explanation of hypertension as kidney function was normal, and no steroids were used. Thus, no definite cause of hypertension in this patients group other than endothelial dysfunction (confirmed by elevation of endothelial dysfunction markers ADMA level).

Limitations: The sample size is rather limited. Despite the low number of patients, we obtained excellent results. Moreover, the pilot study data will be used as a basis for sample size calculations in future epidemiological studies.

Flow-mediated dilatation (FMD), which is the standard method used to non-invasively assess endothelial cell reactivity will be incorporated in future studies [80,81,82,83].

## 5. Conclusions and Future Directions

Autoimmune diseases affect vascular function due to the chronic inflammatory process. The inflammation affects endothelial cells, followed by a disturbance and change in cytokines and chemical mediators (vasoconstrictor and vasodilator), and leads to changes in the homeostasis of the vascular system. Based on our results, we speculate that nontraditional risk factors, in the form of chronic inflammation, initiate or potentiate atherosclerosis through the traditional risk factors (e.g., hypertension). Hypertension is a major traditional risk that is considered an early sign in the process of atherosclerosis [84]. The elevated blood pressure leads to injury and remodeling of the vascular system, and due to the continuous inflammatory process that accompanies autoimmune diseases, the vicious circle persists. Figure 1 summarizes the relationship between autoimmune diseases and vascular function.

We recommend awareness and early detection of arterial hypertension in rheumatic autoimmune diseases. Future studies are needed to confirm the relationship between hypertension and endothelial dysfunction in rheumatoid arthritis and primary Sjorgren syndrome. Finally, we recommend that patients with other autoimmune diseases such as systemic lupus (especially as the kidney plays an important role in the (patho-)physiology of hypertension and a great majority of SLE patients have accompanying hypertension) and scleroderma should be included in future studies.

## Figures and Tables

**Figure 1 jcm-10-04067-f001:**
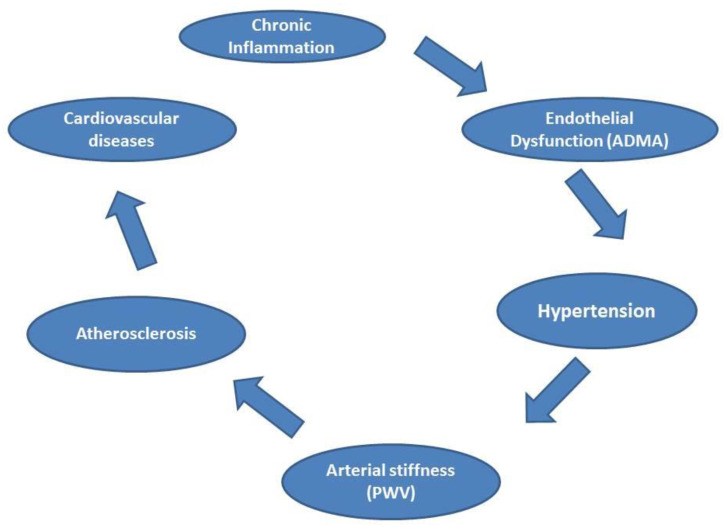
The relationship between autoimmune diseases and vascular function. Legend: asymmetric dimethylarginine (ADMA), pulse wave velocity (PWV).

**Table 1 jcm-10-04067-t001:** General data for the participants. Data are shown as mean ± standard deviation. Legend: RA: rheumatoid arthritis; PSS: primary Sjögren syndrome.

Variable	Control (*n* = 10; 8 Females)	RA (*n* = 10; 8 Females)	PSS (*n* = 10; 10 Females)
Age	Mean 50 ± 9 years	Mean 55.3 ± 7 years	Mean 52.2 ± 8 years
Smoking	*n* = 3	*n* = 1	*n* = 0
Weight	72.5 ± 9 kg	79 ± 12 kg	57.5 ± 16 kg
Creatinine	0.79 mg/dL	0.82 mg/dL	0.78 mg/dL
SBP	128 ± 10 mmHg	148 ± 16 mmHg	135 ± 16 mmHg
DBP	67 ± 5 mmHg	77 ± 8 mmHg	72 ± 8 mmHg
MBP	87 ± 7 mmHg	101 ± 10 mmHg	93 ± 10 mmHg
ADMA	0.38 ± 0.05 µ mol/L	0.45 ± 0.06 µ mol/L	0.43 ± 0.06 µ mol/L
PWV	8.69 ± 1.78 m/s	10.2 ± 2.31 m/s	8.74 ± 1.64 m/s
CRAE	151 ± 21 µm	138 ± 6 µm	146 ± 17 µm
CRVE	215 ± 23 µm	215 ± 17 µm	219 ± 33 µm

**Table 2 jcm-10-04067-t002:** Statistical differences between diseases group and control group (*p*-value). Legend: RA: rheumatoid arthritis; PSS: primary Sjögren syndrome.

Variant	RA	PSS	RA + PSS
SBP	0.007	0.34	0.02
DBP	0.01	0.19	0.01
MBP	0.01	0.24	0.02
ADMA	0.02	0.06	0.02
PWV	0.07	0.48	0.18
CRAE	0.12	0.58	0.21
CRVE	0.94	0.74	0.80

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
