# Peer review of "A Pilot Study: Hypertension, Endothelial Dysfunction and Retinal Microvasculature in Rheumatic Autoimmune Diseases"

_jcm, 2021, doi:10.3390/jcm10184067_

Round 1
Reviewer 1 Report
Endothelial disfunction is one of the major mechanisms leading cardiovascular disease development. The paper “A Pilot Study: Hypertension, Endothelial Dysfunction and Retinal microvasculature in Rheumatic Auto-Immune Diseases” focus on the potential abnormalities which could be found in the patients with autoimmune rheumatic diseases.The study is based on the limited group of the patients which was compared with 10 healthly control subjects. Up to now there is very poor knowledge about the endothelial disfunction as well as abnormalities linking inflammatory autoimmune diseases and atherosclerosis so the paper is valid from the clinical point of view.
From the reviewer point of view some questions should be answered:
- How many of the patients with arterial hypertension received medical treatment before entering the study? What kind of drugs ? – does it influence in the results? (no data about medication presented – but according to the method they were collected)
- Did the patients received anti-inflammatory, immunosuppressive treatment ? What kind? Any correlation with the results? Did the patients received cholesterol lowering therapy?
- Some of the patients were smokers ? what about their results ?
- Figure one suggest that endothelial disfunction can lead to the hypertension and than arterial stiffness and than to atherosclerosis. I think it is very simplified approach, we know that there are several .links between the endothelial disfunction as well as atherosclerosis development – I this this should also be included in the proposed figure 1.
- One of the important tests (approved for endothelial cell reactivity evaluation) is FMD test which was not used in the study for the clinical evaluation of the endothelial disfunction. It is worth to mention this test in the discussion as it could be used for the endothelial cell function evaluation in the further studies on this topic.
Reviewer 2 Report
This is a very interesting field of study. Nevertheless, the current study has a very small and diverse patient population that does not allow for safe conclusions. Furthermore, lacks any novelty concerning previous studies in the field.
several methodological flaws:
1. The study sample is too small and additionally includes divergent forms of rheumatic diseases leading to multiple statistical comparisons. The authors have not performed a statistical power analysis to determine the appropriate sample size.
2. Vascular assessments have inter- and intra-rate variability, that is not being mentioned by the authors. They also do not mention how many investigators performed each measurement.
3. To draw safe conclusions, analysis needs to be performed in multivariate models that have not been assessed by the authors and are not robust in such small sample sizes
4. Based on the above the authors discussion and conclusions cannot be based on the available data.
Reviewer 3 Report
Current version of manuscript is quite poor.
Indeed, POINTs of WEAKNESSES are larger than strenghts.
Nowhere, if we want to save something I think that only METHODs might be considerable as quite good.
Whilst, in my opinion general background and/or discussion have to be consistently improved before considering acceptance.
By starting with a moderate revision of English (form and/or several sentences to be re-written).
In order to try substantial ameliorating, I'd like to suggest some following readings:
- Curr Vasc Pharmacol. 2020;18(6):566-579. doi: 10.2174/1570161118666200127142936.
- Atherosclerosis. 2015 Jul;241(1):259-63. doi: 10.1016/j.atherosclerosis.2015.03.044.
- Clin Exp Rheumatol. 2014 May-Jun;32(3):361-8.
Moreover, the choice of comparison between RA and PSS may be an hazard (probably) because of RA is a paradigmatic and quite too specific rheumatic disease. Then, the authors could be better define a very grey zone by analyzing and adding suggested articles and/or other similar.
Finally, I think quality of tables and chart must be checked.
Best regards.
Round 2
Reviewer 2 Report
The authors have responded to reviewer's comments without making substantial changes to their manuscript.
Reviewer 3 Report
I have much appreciated the efforts of authors in modifying paper following reviewers' suggestions.
Mt
This manuscript is a resubmission of an earlier submission. The following is a list of the peer review reports and author responses from that submission.